# The Driving Mechanism of Urban Land Expansion from 2005 to 2018: The Case of Yangzhou, China

**DOI:** 10.3390/ijerph192315821

**Published:** 2022-11-28

**Authors:** Lin Meng, Wentao Si

**Affiliations:** School of Public Administration, Shandong Normal University, Jinan 250014, China

**Keywords:** urban land expansion, supply and demand theory, path analysis, driving mechanism, Yangzhou

## Abstract

The driving mechanism of expansion is the basis for policymaking in urban land management and control. In this study, Yangzhou city in China was used as an example. Based on the supply and demand theory of land, a framework was constructed to analyze the driving factors behind urban land expansion. Path analysis was used to determine the direct and indirect factors driving urban land expansion. The results showed the following: (1) Urban land in Yangzhou city expanded 32,831.10 hm^2^ from 2005 to 2018, mainly in terms of farmland. This rapid expansion arose from the contraction of rural residential areas, threatening ecological spaces such as water bodies. (2) Path analysis indicated that fixed-asset investment, the added value of secondary and tertiary industries, urban population, distance to the city center, and farmland area are the direct factors driving urban land expansion. Land finance, technology input, and transportation ability indirectly influence urban land by affecting other driving factors. (3) Increasing the level of urban land-use intensification, properly adjusting economic development goals, reasonably controlling the population, exploring the retention of construction land in city centers, and optimizing farmland and basic farmland plans need to be considered in the management and control of urban land expansion. Furthermore, to guide the orderly expansion of urban land, improving land management systems, promoting scientific and technological progress, and scientifically designing transportation land are necessary strategies.

## 1. Introduction

Since the reform and opening-up of China, China has experienced large-scale urban expansion, due to rapid industrialization and urbanization progress. In 1978, there were only 193 cities and 2000 small towns in China. From 1978 to 2002, China implemented the policy of transforming townships into towns and counties into cities. From 1992 to 2013, China experienced a boom in the construction of development zones and new urban areas, along with the reshaping of counties and cities into districts. From 2002 to 2018, metropolitan areas, urban agglomeration, and urban belts began to develop in clusters. Within this process, the scale of urban land in China has expanded. The overall urban pattern has been characterized by agglomeration [1]. In recent years, there has been a phenomenon of local diffusion. The cities and towns around the metropolis, second-tier cities, and cities in the central region are accelerating their development. The city is a complex giant system. The process of urban expansion may result in the convergence of various production factors, thus driving the overall development of the region. Urban land is the facilitator of all social and economic activities in the urban system. Whether urban land expands in an orderly way is related to the healthy development of urbanization. China’s urban land use expanded from 11,608km^2^ in 1990 to 56,076 km^2^ in 2018, with an average annual growth rate of 5.79%. This shows that Chinese cities are developing at an unprecedented speed, and many local governments have used urban development as an important assessment indicator [2]. The continuous expansion of urban land has laid the foundation for more population, capital, and industrial agglomeration. However, while creating social prosperity, it has also caused a series of problems, such as farmland loss and environmental pollution [3]. Urban land is one of the most important indicators in the overall land-use plan, which could guide urban development and control expansion [4,5]. It is vital to study the performance of the driving mechanism of urban land expansion in creating management and control policies for urban land, as well as in improving the management and control system for indicators in land-use design [6].

The current study on the driving mechanism of urban land expansion is based on driving factor selection and driving force measurement. To select the driving factors, a driving mechanism index system was constructed mainly from a society–economy–nature perspective [7]. The cost–benefit theory [8] and internal–external forces [9] are other perspectives that have been employed by other researchers. 

The driving factors in urban land expansion have usually been calculated using a geologic model [6,9], multiple regression analysis [10,11], or a logistic model [12,13]. As it is a special type of product, the enlargement of urban land is restricted by the supply and demand relationship. However, only a few studies have discussed the driving mechanism of urban land expansion from the supply and demand theory perspective [14,15]. Meanwhile, the interaction among the driving factors along with their direct and indirect mechanisms remain unclear. This lack of understanding of the driving mechanism and its pathway has led to incomplete management and control policies in urban land expansion.

Yangzhou city (N 32°15′–33°25′, E 119°01′–119°54′) is a typical ecological city in eastern China, located in the middle part of Jiangsu province, where the north shore of the downstream Yangtze River and the south end of the Yangtze–Huai Plain are found. As an important node city in the Yangtze River Delta economic zone, Yangzhou has a highly developed transportation system, connecting the south of Jiangsu, the north of Anhui, and the north of Jiangsu with their surrounding areas (Figure 1).

Yangzhou has achieved rapid social and economic development since the implementation of the third revision of land utilization planning. The GDP has increased from CNY 92.202 billion to CNY 669.643 billion, along with a rapid increase (40.92%) in urban land expansion. This rapid urbanization poses a great threat to regional food safety and ecological safety. Under the demands of the current national land space plan, the contradictions among economic development, ecological protection, and other goals have become increasingly severe. In this study, Yangzhou city in China was used as an example. Based on the supply and demand theory of land, a framework was constructed to analyze the driving factors behind urban land expansion. Path analysis was used to determine the main factors that drive the expansion of urban land, while direct and indirect pathways were analyzed to address the driving mechanism. This study could provide a reference for the development of management and control policies in urban land expansion.

## 2. Framework Analyzing the Driving Factors of Urban Land Expansion Based on Supply and Demand Theory

### 2.1. Theoretical Analysis of the Influence of the Supply–Demand Relationship on Urban Land Growth

Product price is determined by supply and demand, and the supply–demand relationship is the basis of the market mechanism [14]. In the market economy, as one of the production factors, land goes into the market in the form of a “special product”, inevitably influenced by the supply–demand relationship. Its spatiotemporal configuration must follow the inherent law of supply–demand theory [16,17]. Urban land expansion is restricted by regional supply ability and influenced by social–economic development. Urban land is the dynamic equilibrium in the interaction between land supply and actual demand [5]. The natural supply and economic supply of land resources are the key factors restricting the development and utilization of urban land. City development is a specific use of urban land. Thus, the economic supply of urban land is based on the inelastic natural supply, while being restricted by resource endowment, technology, location and transportation, and land-use intensity [18,19]. Meanwhile, land resource configuration may not spontaneously operate by following the market, urban land supply will be restricted by the macro-regulations that are indicated by the government [20,21]. The government holds the proprietorship of urban land and the development rights of collective land, and the policy system directly determines the economic supply of urban land. With rapid social–economic development, an increasing population, and economic development, there is a need for more living and production space, leading to a continuous increase in the demand for urban land. The limited supply of urban land and increasing urban land demand determine the regional urban land expansion pattern through the regulation effect of market mechanisms. In this study, we start from the land supply and demand perspective, constructing a theoretical analysis framework of the driving factors in urban land expansion (Figure 2).

#### 2.1.1. Supply Elements

(1) Natural endowment feature. The regional natural endowment feature determines the natural supply of land, and further influences the flexibility of the economic supply of land [22]. The best urban land is always located in the plain areas. The higher the elevation and slope, the lower the land supply capability [23]. The topography of Yangzhou influenced the urban land supply suitability due to its special features, displaying a fan-shaped slope from west to east. Urban land expansion was mainly fulfilled by occupying farmland, as farmland was the main supply source of urban land [24,25]. Thus, in this study, slope, elevation, and farmland area were selected as indicators to evaluate the regional endowment feature.

(2) Technology level. According to the Solow growth model, the technology level can increase the marginal returns of work, thus decreasing the importance of land, having a positive effect on capital investment in unit land, finally enhancing the economic supply of land resources [26]. Investment in technological upgrading, which represents the technology level, was selected as an indicator to evaluate the regional technology level.

(3) Location. The distance to city center is the most important location factor influencing urban land expansion. A short distance to the city center leads to a stronger intensity of social–economic activity, thus increasing the possibility of developing land for the city [27]. Developed transportation can significantly promote the acceptance of market influence and decrease the transfer cost, which will increase the economic supply of land [28]. Thus, the distances to city center and transportation accessibility were selected as indicators to evaluate the location condition.

(4) Land-use intensity. Based on the Harrod–Domar model [29], the land-use intensity level reflects the substitution relationship between land and capital. If the other parameters are fixed, only the contribution of land and capital to the economy is considered, capital could stimulate the economy, and land and capital could be replaced by technology. Thus, average fixed-asset investment was selected as an indicator to evaluate land-use intensity [30].

(5) Policy system. The local government is the administrator of the land market, as well as the oligopolist in the primary land market [31]. This double identity resulted in the local government generally increasing its financial income through land leasing. In the 1990s, China implemented tax sharing system reform and established a tax sharing budget management system. In China, on the basis of a reasonable division of the scope of powers of governments at all levels, the budget revenues of governments at all levels were mainly divided according to taxes. The budgets of governments at all levels were relatively independent and had a clear responsibility for balancing. The financial difference between governments at all levels was adjusted through the transfer payment system from the central government to the local government, or from the local government at the next higher level to the local government at the next lower level. The tax sharing budget management system clarified the division of central and local fiscal expenditure, central and local fiscal revenue, and the intergovernmental fiscal transfer payment system. After tax reform, the government try to maximize disposable income. In order to relieve local financial stress, local governments try to facilitate the “land to money, money to land” circle through land condemnation and transfer. Land finance has become an important way to increase local financial income and local government achievements [32]. However, there are differences in the land regulation policies and supervision efforts of local governments in different regions, resulting in differing dependence on “land finance”. Therefore, there are differences in behavior choices in urban land supply. Different behavior will be selected in urban land supply due to the difference in the dependency on land finance [33]. The income comes from land transfer is the main source of land financial benefit [34]. Thus, land transfer income was selected as indicator to evaluate the policy system that local governments adopted for urban land expansion. 

#### 2.1.2. Demand Elements

(1) Population increase. Population growth leads to an increase in consumption demand, which is the fundamental factor affecting land demand [35,36]. Urban population growth results in pressure to increase the existing land-use space. Therefore, more production and living spaces need to be added to meet the needs of population growth, which leads to a rapid expansion of the urban construction land scale [37]. Thus, the urban population increase amount was selected as indicator to evaluate population increase.

(2) Economic development. Regional economic development needs investment in a production element. The demand for urban land is influenced by economic development because urban land is the main resource for urban construction and industrial development. The characteristic of total economic development influences the demand for urban land, further driving area change [38]. Urban land is the main resource of the secondary and tertiary industries. Thus, the added value of the secondary and tertiary industries was selected as indicator to evaluate the economic development level.

### 2.2. Path Analysis on the Influence Mechanism of Urban Land Expansion

Path analysis is one type of multivariate statistical analysis technology used to study the interaction between explanatory variables, as well as the influence of explanatory variables on explained variables [39]. Direct path is used to address the direct effect of one specific explanatory variable on an explained variable, while indirect path is used to address the indirect effect of other explanatory variables on explained variables [40]. Assuming that independent variable *y* was influenced by multiple factors$\;x_{1}$,$x_{2}$,…,$x_{m}$, and each factor was linearly correlated with *y*, we can construct the following multiple regression equation:(1)$$y=b_{0}+b_{1}x_{1}+b_{2}x_{2}+\ldots +b_{m}x_{m}$$
where, if we divide the correlation coefficient $r_{iy}$ between dependent variables $x_{i}$(i = 1,2,…,*m*) and independent variable y into the direct effect of $x_{i}\;\mathrm{to}\;y$ (direct path coefficient), and the indirect effect of $x_{i}$ to y that acts through other dependent variables (indirect path coefficient), the relative importance of each variable could be directly compared.

The normal equation of path coefficient $p_{01}\ldots p_{0m}$ could be drawn as below:(2)$$\left\{\begin{array}{c} p_{1y}+r_{12}p_{2y}+\ldots r_{1m}p_{my}=r_{1y}\\ r_{21}p_{1y}+p_{2y}+\ldots r_{2m}p_{my}=r_{2y}\\ \ldots \ldots \ldots \ldots \ldots \ldots \ldots \ldots \ldots \ldots \ldots \ldots \ldots \ldots ..\\ r_{m1}p_{1y}+r_{m2}p_{my}+\ldots r_{mm}p_{my}=r_{my} \end{array}\right.$$
where $r_{ij}$ is the correlation coefficient of $x_{i}$ to $x_{j}$; $r_{iy}$ denotes the correlation coefficient of $x_{i}$ to y, i.e., the integrated influence of $x_{i}$ on y; $p_{iy}$ represents direct path coefficient, i.e., the normalized partial correlation coefficient of $x_{i}$ to y, which shows the direct effect of $x_{i}$ on y.
(3)$$\;\;q_{ij}=r_{ij}p_{iy}$$
(4)$$\;\;q_{iy}=\underset{i\neq j}{\sum }r_{ij}p_{iy}$$

In Equation (3), $q_{ij}$ is the direct path coefficient between paired variables, indicating the indirect effect of $x_{i}$ on *y* acting through $x_{j}$. In Equation (4), $q_{iy}$ is the indirect path coefficient, showing the total indirect effect of $x_{i}$ on *y* acting through all other explanatory variables.

## 3. Data Source and Data Processing

Yangzhou was selected as the study area. The research period was 2005–2018. At the same time, townships (87 townships) were used as the basic research units. Urban land data from 2005, 2013, and 2018 were obtained from the Chinese Academy of Sciences Geography Science and Resource Institute website. The 2009 urban land data were derived from the Yangzhou Statistical Yearbook and the Statistical Yearbook of each district. Geospatial data, such as distance to city center, elevation, and slope data, as well as land classification data were also obtained from the Chinese Academy of Sciences Geography Science and Resource Institute website. Investments in industrial conversion, road mileage, and other social–economic data were from the Yangzhou Statistical Yearbook and the Statistical Yearbook of each district. Table 1 shows the source of each variable. 

## 4. Results Analysis

### 4.1. Urban Land Expansion Scale 

From 2005 to 2018, the scale of urban land use in Yangzhou showed an upward trend, with a total expansion area of 32,929.60 hm^2^. The expansion scale of urban land in 2005–2013 was 17,720.98 hm^2^. On the district (county, city) scale, Hanjiang had the largest expansion in this period, with a total expansion area of 4267.48 hm^2^, accounting for 24.08% of the total. Baoying had the smallest expansion, accounting for 9.85% of the total. From 2013 to 2018, the total expansion area of urban land in Yangzhou was 17,750.62 hm^2^. Jiangdu had the largest expansion in this period, with a total expansion area of 3552.70 hm^2^, accounting for 23.36% of the total. Guangling had the smallest expansion, accounting for 9.20% of the total. The total urban land expansion area was 32,831.10 hm^2^ from 2005 to 2018 in Yangzhou. The newly increased area of urban land was mainly located in the city center and in the rural areas of Jiangdu, Yizheng, Gaoyou, and Baoying, of which, the urban land expansion in Hanjiang was the highest, accounting for 22.7% of the total expansion area. The lowest was Yizheng, accounting for 12.88% of total expansion area. In summary, although the main expansion areas (counties) of urban land in different periods were different, the scale of urban land in the study area took the central urban area as the main expansion core area, and the growth plates were concentrated near the central urban area, with obvious clustering distribution characteristics (Figure 3, Table 2).

From an urban land expansion source perspective, construction land is the most important source, followed by agricultural land and unused land, accounting for 47.25%, 46.02%, and 6.73%, respectively (Table 2). 

### 4.2. Path Analysis of Urban Land Expansion

The multilinear index must be removed before conducting a path analysis. The driving factors that influenced the independent variables were selected for subsequent analysis. Stepwise regression analysis was selected to screen the driving factors of urban land expansion, as it is an important method for constructing the optimal regression equation under the condition that the dependent variables have collinearity. The standard regression coefficient is defined as the direct path coefficient for each driving factor. 

The stepwise analysis and path analysis results (Table 3) showed that the driving factors of urban land expansion were sorted according to their contribution from high to low. The order was as follows: added value of secondary and tertiary industries (x_8_), urban population (x_7_), technology investment (x_2_), distance to city center (x_3_), average fixed-asset investment (x_5_), land finance scale (x_6_), transportation accessibility (x_4_), and farmland area (x_1_) (Table 3, Figure 4).

#### 4.2.1. Recognition of Direct Factors

The direct driving factors of urban land expansion were sorted according to the direct path coefficient, from high to low. The order was: x_5_, x_8_, x_7_, x_3_, x_1_, x_6_, x_2_, and x_4_, of which, the direct path coefficients of x_5_, x_8_, x_7_, x_3_, and x_1_ were higher than their indirect path coefficients, thus they were defined as direct factors (Table 3, Figure 4). 

#### 4.2.2. Recognition of Indirect Factors

The indirect driving factors of urban land expansion were sorted according to the indirect path coefficient, from high to low. The order was: x_6_, x_2_, x_8_, x_7_, x_3_, x_4_, x_1_, and x_5_, of which, the indirect path coefficients of x_6_, x_2_, and x_4_ were higher than their direct path coefficients, influencing urban land expansion through other driving factors, thus they were defined as direct factors (Table 3, Figure 4). 

## 5. Discussion

### 5.1. Urban Land Expansion Scale 

In the agricultural land, farmland was the main source of urban land expansion, accounting for 40.55%. In the construction land, rural residential land was the main source, accounting for 40.40%. This may be due to the implementation of rural residential management policies, such as “Million hectares of fertile farmland” and “Link the increase and decrease of urban-rural construction land”, which enabled the spatial displacement and development rights to transform between urban and rural construction lands. In the unused land, water areas were the main source of urban land, accounting for 6.01%. In summary, farmland was the main source of urban land expansion, while the decrease in residential land in rural areas was another important source supporting the rapid increase in urban land. Apart from this, water space was occupied by urban land expansion, accounting for 6.45% of the total expansion area. However, ecological spaces, such as water areas, were threatened in the urban land expansion process (Table 2).

### 5.2. Path Analysis of Urban Land Expansion

#### 5.2.1. Recognition of Direct Factors

The direct path coefficient of x_5_ was −0.32, directly inhibiting urban land expansion. It was indicated that urban land expansion could be effectively inhibited through land-use intensity enhancement and using capital to replace land investment. The direct path coefficient of x_8_ was 0.30, directly promoting urban land expansion. The secondary and tertiary industries developed rapidly due to the new economic development strategy, which was based on an advanced manufacturing and service industry. Finally, this led to a rapid increase in urban land expansion demand. The direct path coefficient of the urban population (x_7_) was 0.28. As the urban population increased, the urban land demands for residence, industry, and entertainment increased rapidly, leading to continuously increased urban land [20]. The direct path coefficient of x_3_ was −0.22. This indicated that the city center was the important urban land expansion source. The shorter the distance to city center, the higher the possibility that the land will transfer to urban land. The direct path coefficient of x_1_ was −0.18, directly inhibiting urban land expansion. This may be due to the strict farmland protection policy, which restricted the transformation of farmland to urban land. Another possible reason may be the long distance from the intensive farmland area to the city center, as the low urban population in those areas indirectly inhibits urban land expansion (Table 3).

#### 5.2.2. Recognition of Indirect Factors

The direct path coefficient of x_6_ (−0.12) was smaller than the indirect path coefficient (0.41), defined as an indirect factor indirectly promoting urban land expansion (Figure 5a). The indirect influence was acting through x_8_, x_7_, and x_3_. The indirect path coefficients between each paired variable were 0.11, 0.08, and 0.07, respectively. Expanding urban land areas by transferring collective land into state-owned land, and by recycling full due land, illegal land, and unused land were the two main ways the government obtained benefits from land transfer. The local government preferred to use the first method because it was easy to implement, and could maximize the benefit of land transfer [41]. This resulted in rapid urban land expansion. The region has a large population, a good location, and a developed economy, which was attractive to investors. Meanwhile, the government focused on this region as a means to obtain greater financial income. This indicates that the local government directly and indirectly promoted urban land expansion as they pursued land finance and administrative achievement (Figure 5).

The indirect path coefficient of technology investment (0.29) is larger than the direct path coefficient (0.08), indirectly promoting urban land expansion (Figure 5b). The indirect influence was achieved through x_8_, x_5_, and x_7_. The indirect path coefficients between each paired variable were 0.16, 0.13, and 0.08, respectively. One possible reason for the indirect promoting effect may be that the technology investment was mainly concentrated in the national economic–technological development area located in the city center. Economic growth is one of the driving forces of urban land expansion. Meanwhile, the region with advanced technology was followed by developed industry, highly intensive but still attracting a large amount of labor, thus promoting urban land expansion (Figure 5).

The indirect path coefficient of transportation accessibility (−0.10) was larger than the direct path coefficient (−0.07), indirectly inhibiting urban land expansion (Figure 5c). The indirect influence acted through x_1_, x_6_, and x_7_. The indirect path coefficients between each paired variable were −0.06, −0.06, and 0.05, respectively. The urban land expansion area was concentrated near the city center and the central town beside the highway. Many new roads had been constructed outside the city center and the central town in the previous years, thus the transportation accessibility was greatly improved. The large farmland area and low land financial income inhibited urban land expansion in those regions. Meanwhile, improved transportation accessibility in the outer suburban districts promoted population movement, which was beneficial for increasing the population, indirectly promoting urban land expansion. However, considering multiple factors, transportation accessibility poses an indirect inhibitory effect on urban land expansion (Figure 5).

### 5.3. Policy Suggestions

(1) Average fixed-asset investment, the added value of secondary and tertiary industries, urban population, distance to the city center, and farmland area are important direct factors that drive urban land expansion. The integrated and direct effect of the first four factors plays a more important role. Indirect factors often act through them to indirectly influence urban land expansion. Based on these results, we propose the following suggestions. In order to slow down the speed of urban land expansion, improving land-use intensification and promoting the three-dimensional development of urban land could be considered. Moderately adjusting the economic development goals, maintaining an appropriate economic development speed, optimizing the industrial structure, and promoting the development of a modern service industry could decrease the dependency of economic development on urban land. The demand for urban land could be reduced by controlling the population. Considering the location factor could help in exploring the retention of construction land in the city center. Combining the “Million hectares of fertile farmland” and “Link the increase and decrease of urban-rural construction land” policies, optimizing the farmland and basic farmland design, and assigning part of the farmland to urban development could control urban land expansion through ecological isolation.

(2) Land finance scale, technology investment, and transportation accessibility directly affect urban land expansion, as well as indirectly influencing urban land expansion through other factors. However, the indirect effects have an obvious lagging effect. To break the government monopoly on land supply, land management policy should be improved. Several strategies could be attempted: (1) Improve the method of land condemnation by restricting local government land transfers and initiate the reuse of urban land storage in the regions that have a better location and economy, and a larger population. (2) Accelerate financial system reform to adjust land transfer policies. (3) Find an effective way to overcome the difficulties with land finance. Construct a complete local tax system by reasonably levying municipal governments to guarantee a steady increase in local income. Meanwhile, to reduce the demand for urban land, industry structures should be updated through technology processing. To achieve this target, the following strategies could be tried: (1) continue to improve the level of regional technology and reasonably allocate land for high-tech industries; and (2) move some high-tech industries to city centers, and reduce the urban land supply stress in city centers. Finally, transportation land should be designed in scientific way to avoid repeated construction. The total amount of transportation land should be controlled to accomplish orderly urban land expansion.

## 6. Conclusions

The total urban land expansion area was 32,929.60 hm^2^ in Yangzhou from 2005 to 2018. The newly increased urban land was mainly located in the city center and in the rural areas of Jiangdu district, Yizheng, Gaoyou, and Baoying. The expansion in urban land was dominated by farmland, which accounted for 40.55% of the total expansion area. Meanwhile, a decrease in residential land in rural areas was another important source of land for urban land expansion, accounting for 40.40% of the total expansion area. Furthermore, water spaces were occupied in urban land expansion, accounting for 6.01% of the total expansion area.

The driving factors of urban land expansion were sorted according to their contribution, from high to low. The order was as follows: added value of secondary and tertiary industries, urban population, technology investment, distance to city center, average fixed-asset investment, land finance scale, transportation accessibility, and farmland area. Of these, average fixed-asset investment, added value of secondary and tertiary industries, urban population, distance to city center, and farmland area were the direct factors. Land finance scale, technology investment, and transportation accessibility were the indirect factors, which indirectly influenced urban land expansion through other driving factors. Land finance scale promoted urban land expansion through the added value of secondary and tertiary industries, urban population, and distance to the city center. Technology investment assisted urban land expansion through the added value of secondary and tertiary industries, average fixed-asset investment, and urban population. Transportation accessibility inhibited urban land expansion through farmland area, land finance scale, and urban population.

Direct and indirect driving factors should be considered in the management and control of urban land expansion policymaking. From a direct factor perspective, increasing the level of urban land-use intensification, properly adjusting economic development goals, reasonably controlling the population, tapping into the power of the retention of construction land in city centers, and optimizing farmland and basic farmland planning need to be considered in the management and control of urban land expansion. Moreover, from an indirect factor viewpoint, to guide the orderly expansion of urban land, improving land management systems, promoting scientific and technological progress, and scientifically designing transportation land are necessary strategies.

## Figures and Tables

**Figure 1 ijerph-19-15821-f001:**
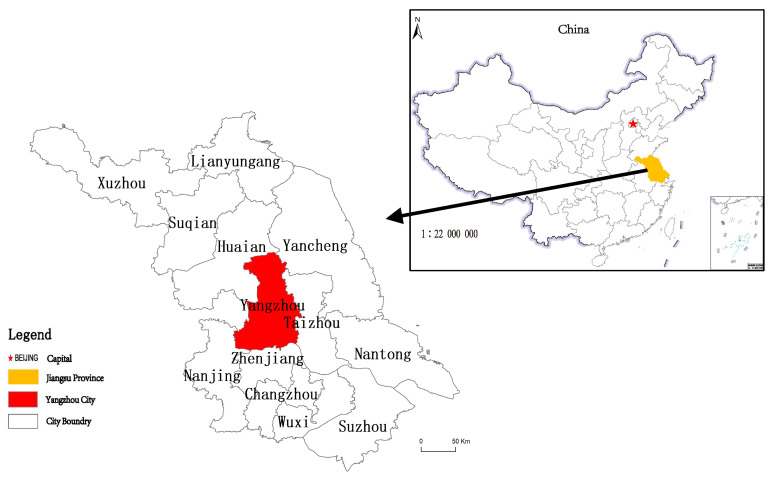
The location of the study area in China.

**Figure 2 ijerph-19-15821-f002:**
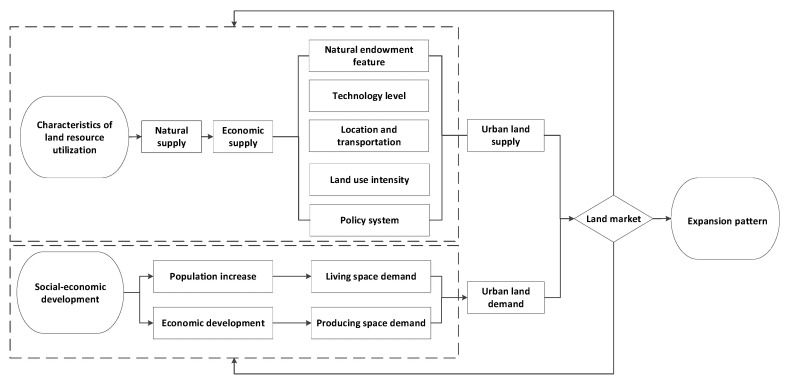
The theoretical framework of supply/demand-induced urban land expansion.

**Figure 3 ijerph-19-15821-f003:**
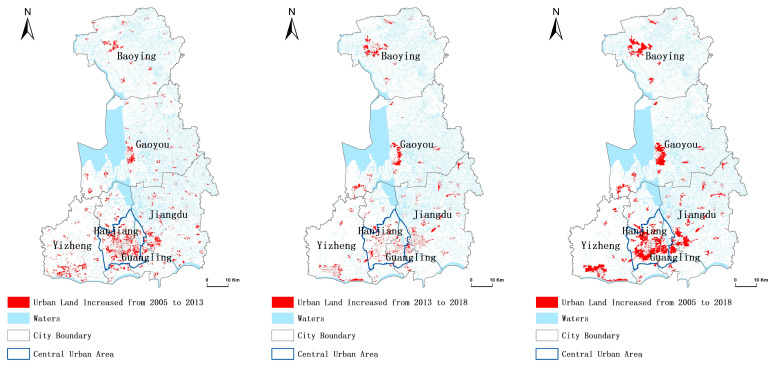
Patch distribution of urban land expansion in Yangzhou.

**Figure 4 ijerph-19-15821-f004:**
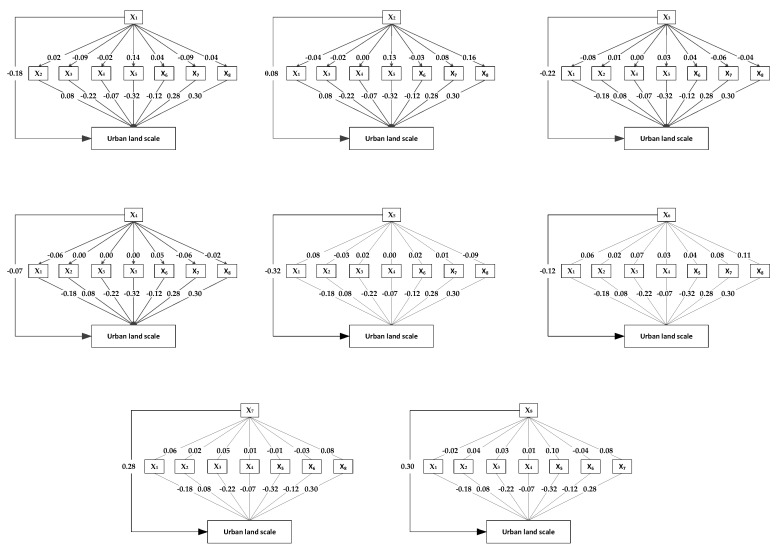
Path analysis of urban land expansion. Note: x_1_: farmland area, x_2_: technology investment, x_3_: distance to city center, x_4_: transportation accessibility, x_5_: average fixed-asset investment, x_6_: land finance scale, x_7_: urban population, x_8_: added value of secondary and tertiary industries.

**Figure 5 ijerph-19-15821-f005:**
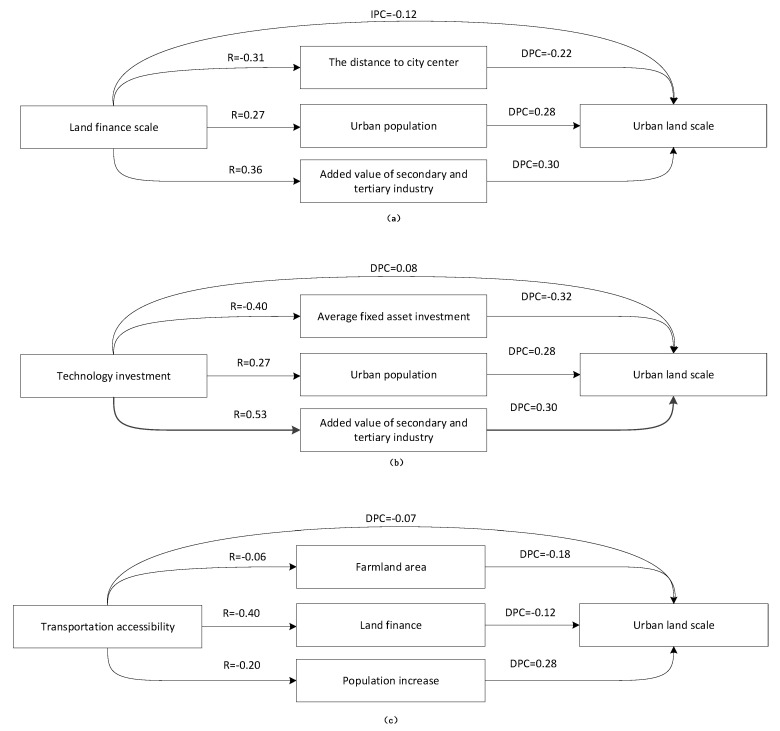
Action path of indirect path coefficient (correlation coefficient: R; direct path coefficient: DPC). Note: (**a**): The direct and indirect influence path of land finance scale on urban land scale. (**b**): The direct and indirect influence path of technology investment on urban land scale. (**c**): The direct and indirect influence path of transportation accessibility on urban land scale.

**Table 1 ijerph-19-15821-t001:** The driving factors of urban land expansion.

Driving Factor		Variable	Variable Description	Source
Supply factors	Natural endowment feature	Farmland area	Current farmland area (hm^2^)	Yangzhou Statistical Yearbook (2010, 2014, 2016, 2019)/Statistical Yearbook of each district (2010, 2014, 2016, 2019)/Chinese Academy of Sciences Geography Science and Resource Institute (http://www.resdc.cn/DataList.aspx) (accessed on 20 June 2022)
		Slope	Slope of land (°)	Chinese Academy of Sciences Geography Science and Resource Institute (http://www.resdc.cn/DataList.aspx) (accessed on 20 June 2022)
		Elevation	Altitude (m)	Chinese Academy of Sciences Geography Science and Resource Institute (http://www.resdc.cn/DataList.aspx) (accessed on 20 June 2022)
	Technology level	Technology investment	Investment in industrial conversion (10^4^ CNY)	Yangzhou Statistical Yearbook (2010, 2014, 2016, 2019)/Statistical Yearbook of each district (2010, 2014, 2016, 2019)
	Location and transportation	Distance to city center	Average distance from urban land to city center (m)	Chinese Academy of Sciences Geography Science and Resource Institute (http://www.resdc.cn/DataList.aspx) (accessed on 20 June 2022)
		Transportation accessibility	Road mileage (km)	Yangzhou Statistical Yearbook (2010, 2014, 2016, 2019)/Statistical Yearbook of each district (2010, 2014, 2016, 2019)
	Land-use intensity	Average fixed-asset investment	Fixed-asset investment/urban land area (10^4^ CNY/hm^2^)	Fixed-asset investment: urban land area: Chinese Academy of Sciences Geography Science and Resource Institute (http://www.resdc.cn/DataList.aspx) (accessed on 20 June 2022)
	Policy system	Land finance	Income from land transfer (10^4^ CNY)	Land market data on the websites of natural resources and planning bureaus of all districts
Demand factors	Population increase	Urban population	Population in urban area	Yangzhou Statistical Yearbook (2010, 2014, 2016, 2019)/Statistical Yearbook of each district (2010, 2014, 2016, 2019)
	Economic development	Added value of secondary and tertiary industries	Sum of added value from secondary and tertiary industries (10^4^ CNY)	Yangzhou Statistical Yearbook (2010, 2014, 2016, 2019)/Statistical Yearbook of each district (2010, 2014, 2016, 2019)

**Table 2 ijerph-19-15821-t002:** Source analysis of urban land expansion in Yangzhou (unit: hm^2^).

	District	Baoying	Gaoyou	Guang-Ling	Han-Jiang	Jiangdu	Yizheng	Total
Land Classification								
Agricultural land	Farmland	2326.04	3086.12	1353.99	2213.61	3104.91	1268.15	13352.81
	Garden plot	14.21	37.94	33.20	37.45	47.60	5.05	175.46
	Forest land	65.88	20.35	54.14	59.56	34.69	124.06	358.68
	Others	206.06	385.10	144.58	277.33	112.64	140.50	1266.22
Construction land	Residential area	1245.22	1415.19	2463.70	3743.94	2435.92	1999.83	13303.81
	Transportation and water conservancy	358.69	263.47	227.01	623.86	310.47	423.33	2206.83
	Others	13.11	1.47	1.00	15.39	3.26	14.35	48.59
Unused land	Water area	173.12	459.05	304.72	475.84	424.85	142.69	1980.27
	Mudflat and swamp	0.00	4.13	3.59	4.27	17.19	109.24	138.41
	Others	0.00	36.82	0.97	20.70	25.33	14.69	98.50
	Total	4402.35	5709.65	4586.90	7471.94	6516.86	4241.90	32929.60

**Table 3 ijerph-19-15821-t003:** Path analysis of urban land expansion.

EV	CC	DPC	Indirect Path Coefficient among Variables								IPC
			x_1_	x_2_	x_3_	x_4_	x_5_	x_6_	x_7_	x_8_	
x_1_	−0.159 *	−0.18		0.02	−0.09	−0.02	0.14	0.04	−0.09	0.04	0.02
x_2_	0.369 **	0.08	−0.04		−0.02	0.00	0.13	−0.03	0.08	0.16	0.29
x_3_	−0.330 **	−0.22	−0.08	0.01		0.00	0.03	0.04	−0.06	−0.04	−0.11
x_4_	−0.172 *	−0.07	−0.06	0.00	0.00		0.00	0.05	−0.06	−0.02	−0.10
x_5_	−0.319 **	−0.32	0.08	−0.03	0.02	0.00		0.02	0.01	−0.09	0.00
x_6_	0.294 **	−0.12	0.06	0.02	0.07	0.03	0.04		0.08	0.11	0.41
x_7_	0.474 **	0.28	0.06	0.02	0.05	0.01	−0.01	−0.03		0.08	0.19
x_8_	0.491 **	0.30	−0.02	0.04	0.03	0.01	0.10	−0.04	0.08		0.19

Note: IPC means indirect path coefficient; ** indicates significantly correlated at *p* < 0.01; * indicates significantly correlated at *p* < 0.05; x_1_: farmland area, x_2_: technology investment, x_3_: distance to city center, x_4_: transportation accessibility, x_5_: average fixed-asset investment, x_6_: land finance scale, x_7_: urban population, x_8_: added value of secondary and tertiary industries.

## Data Availability

Not applicable.

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
