# Peer review of "The Driving Mechanism of Urban Land Expansion from 2005 to 2018: The Case of Yangzhou, China"

_ijerph, 2022, doi:10.3390/ijerph192315821_

Round 1

Reviewer 1 Report

This work analysis the driving mechanism of urban land expansion from 2005 to 2018 in Yangzhou, China. The research is of great significance to improve land management system as well as design transportation. Personally, I think the authors have done a lot of work, however, there are some deficiencies that need to be further improved.

Question1#: The results of the fourth part are thin overall, with only five paragraphs to describe.

Question2#: There are some mistakes in the text. For example, GDP in Yangzhou increased from 92.202 to 69.643 (P 2 L58-L59); The urban land expansion in Baoying was lowest is right? Yizheng (4227.22) and Baoying (4402.35) (P 6 L190-L191&P 7 Table 2), the description of this part needs to be confirmed; The form of symbols of x1, x2…x8, is inconsistent.

Question3#: Table 3 does not seem to intuitively reflect the relationship between variables. Could you draw a structural equation model diagram about the mechanism of action and transmission mechanism through the constructed structural equation.

Reviewer 2 Report

It is necessary an extensive editing of English language and style (eg lines 130,137, 140-141, 175 etc). Also Figure 1 is not mentioned in the article. 

On conclusions, it is not needed (1), (2), (3). 

Reviewer 3 Report

The paper is well explained in the text but is lacking of images that would improve the study understanding.

 I suggest to insert the following figures:

1. figure with the territories identification of the different districts

2. maps showing the different land use over time, perhaps at the beginning and at the end of the reference period of the study.

In the introduction, a short note about the economic reform in China could help to understand the impact on the urbanization processing.

Figure 1. The marker of Beijing isn’t well visible

Row 125. What has the tax reform been about?

The main question addressed by the research is what are the driving mechanism of urban expansion in a study case and what are the strategies to improve land management.

The topic is interesting, relevant to the field.

It’s defines a methodology and applies it in a study case.

The methodology as an analysis of the topic is interesting but should be supplemented by figures and maybe diagrams that make the document better readable.

The conclusions are appropriate.

The references are appropriate. Many references are recent too.

The tables are clear except table 3 that is not immediately readable. Is it possible to find a way to better read the relationships between the variables?

Round 2

Reviewer 1 Report

By modifying the wording of the entire article, it can be received.

Reviewer 3 Report

I thank the authors for having accepted my suggestions. I believe that the paper is now clearer and easier to read.

Best regards